# Coastal Scenic Evaluation of Continental Ecuador and Galapagos Islands: Human Impacts and Management Issues

**Carlos Mestanza-Ramón** [1,2,3], **Giorgio Anfuso** [1,*], **J. Adolfo Chica-Ruiz** [1], **Alexis Mooser** [1,4], **Camilo M. Botero** [5,6] and **Enzo Pranzini** [7]

1   Facultad de Ciencias del Mar y Ambientales, Universidad de Cádiz, Polígono Río San Pedro s/n, 11510 Puerto Real, Spain; carlos.mestanza@espoch.edu.ec (C.M.-R.); adolfo.chica@uca.es (J.A.C.-R.); alex.moosr@gmail.com (A.M.)
2   YASUNI-SDC Research Group, Sede Orellana, Escuela Superior Politécnica de Chimborazo, El Coca EC220001, Ecuador
3   Instituto Superior Tecnológico Universitario Oriente, La Joya de los Sachas, Orellana 220101, Ecuador
4   Dipartimento di Scienze e Tecnologie, Università di Napoli Parthenope, 80143 Naples, Italy
5   Grupo Joaquín Aarón Manjarrés, Escuela de Derecho, Universidad Sergio Arboleda, Santa Marta 470001, Colombia; playascol@gmail.com
6   Grupo de Investigación en Sistemas Costeros, PlayasCorp, Santa Marta 470001, Colombia
7   Dipartimento di Scienze della Terra, Università di Firenze, 50121 Firenze, Italy; enzo.pranzini@unifi.it
*   Correspondence: giorgio.anfuso@uca.es; Tel.: +34-956-016-167

**Abstract:** The scenery, safety, facilities, water quality and litter quantities in coastal areas are relevant and determining elements in the choice of a tourist destination. This paper focused on the evaluation of coastal scenic value in 55 and 12 sites respectively located in continental Ecuador and the Galapagos Islands. The information obtained gives public administrators and coastal managers the relevant data to avoid further environmental degradation and suggests measures to improve the present scenic value of tourist destinations. The methodology used was based on the analysis of 26 physical/human factors and applied fuzzy logic analysis and weighting matrices that allowed the sites to be classified into five classes, from Class I (natural areas with superior scenic characteristics) to Class V (poor scenic areas with relevant impact of human interventions). The most attractive beaches were in the Galapagos Islands due to the magnificent physical and environmental characteristics, while the Esmeraldas province presented sites of lower scenic beauty due to the low natural scenic value and the increase of human impacts. In total, 22% (15 out of 67) of the beaches investigated belonged to Class I, 12% (8) to Class II and 15% (10) to Class III. The last two classes included 51% of the beaches (i.e., 34 out of 67), of which 31% (21) was in Class IV and 20% (13) in Class V. Such results provide local managers and planners a solid inventory on coastal scenic characteristics and baseline information for any envisaged subsequent management plan.

**Keywords:** beach; tourism; landscape; protected area; fuzzy logic

## 1. Introduction

The importance of landscape for society has been recognized for a long time and, nowadays, tourism and coastal scenery represent two intimately related realities [1]. Several anthropic and natural factors directly determine and affect the scenic value of a site that, to be properly preserved, needs sound management actions and strategies [1].

Nowadays, human impacts on coastal scenery are essentially linked to 'travel and tourism', which is one of the fastest growing industries in the world. International tourist arrivals were 25 million in 1950 and are estimated to be 1.8 billion by 2030 [2,3]. In 2019, international tourist arrivals were 1.5 billion, corresponding to an increase of 4% with respect to 2018 [3,4]; this is a slower increase with respect to the one (+6%) recorded in 2017 and 2018 due to the global economic slowdown, the uncertainty related to the Brexit, and commercial and geopolitical tensions [3].

The arrival of foreign visitors in Ecuador in 2018 increased by 11% compared to 2017 [3]. Tourism in Ecuador greatly contributes to the national Gross Domestic Product (GDP), both in a direct (2%) and indirect/induced (5%) way and it accounts for about 1 out of 20 jobs, making tourism the fifth largest economic industry of the country [5]. According to the Travel Account of the Central Bank of Ecuador, the income of foreign currency linked to international visitors has grown by 7% between 2013 and 2017 and, in the latter year, tourism was responsible for USD 1633 million entrances, occupying in this way the third place—after revenues linked to banana and shrimp exportations—for foreign income currency between the non-oil goods [6]. Hence, tourism has been identified as one of the most relevant activities to the economic development of the country. Central Government and Regional Administrations have the capacity to influence and take action in the issues that affect this activity through regulations, incentives, promotion or by mitigating negative external influences. Ecuador is no stranger to these actions; the participation of government actors, trade unions, social organizations and communities has generated a great boost to tourism. Tourism is currently considered a State policy and a priority issue on the national political agenda as one of the country's main economic activities [7]. These policies aim to attract a greater number of foreign tourists and to boost the local economy in a sustainable manner. In recent years, the country has maintained a growing economic trend which is important in the region, as evidenced by the growth of its gross domestic product [8]. Although Ecuador is not among the best countries in Latin America, it has been characterized by maintaining an average growth rate of 4.30% compared to the average for the region (South America) of 3.85%. Ecuador's economic growth is due to a series of important decisions on economic income generators, moving from being a country focused on the primary sector to developing the industrial and, especially, service sectors [6].

According to the Ministry of Tourism of Ecuador, 57% of international visitors that enter in the country are essentially interested in tourist activities such as cultural tourism (44%), ecotourism (30%), the Sun, Sand and Sea (3S) tourism (21%), adventure tourism (4%) and other types of tourism (1%) [6]. This was also observed at the international level [9,10]; beaches are a major player in tourist market. Within the Ecuadorian market, Quito was the most important destination in 2017 with 72.3% of international visitors followed by Guayas (51.2%), Santa Elena (31.4%) and Tungurahua (29.8%) provinces. In 2017, the Galapagos Islands recorded 167,051 foreign tourists (that corresponds to an increase of 12% compared to 2016), especially interested in adventure, ecotourism and beach activities [6].

The habit of frequenting the beach dates back to when the wealthy English society started to search for beach and sun [11]. At present, the coastal landscape has received great attention from the 3S tourism researchers [12–15] but, despite this, studies on specific resources related to this kind of tourism are relatively scarce. Several authors [16–18] highlighted water quality, safety, absence of litter, facilities and landscape as most relevant aspects linked to the 3S tourism, the latter being the focus of this research.

The objective of this paper is to evaluate the coastal landscape at 67 beaches on the continental coast of Ecuador and the Galapagos Islands (Figure 1 and Table 1). The study was carried out according to the methodology proposed by Ergin et al. [16], which is based on fuzzy logic analysis and parameter weighting matrices. Coastal scenic evaluation constitutes an extremely relevant tool for coastal knowledge, preservation and future development, as this provides a sound scientific basis for any envisaged coastal management plan. Information recollected in this investigation was cross referenced with the topographic and geological setting of the study area, as well as with existing data on tourist typologies distribution [18] to prevent further environmental degradation, but also to suggest measures to improve the present scenic value of tourist destinations [19–21].

## 2. Materials and Methods

### 2.1. Study Area

Ecuador, with a total surface of 270,670 km$^2$, is one of the smallest countries in South America. From an administrative point of view, the Ecuadorian coastline includes several continental provinces, i.e., Esmeraldas, Manabí, Santa Elena, Guayas and El Oro and the insular province of Galapagos. The present investigation was concentrated in Esmeraldas, Manabí and Santa Elena, and the Galapagos (Figure 1 and Table 1) [22]. The provinces of Guayas and El Oro were discarded because of their high concentration of mangrove ecosystems and negligible number of tourist beaches.

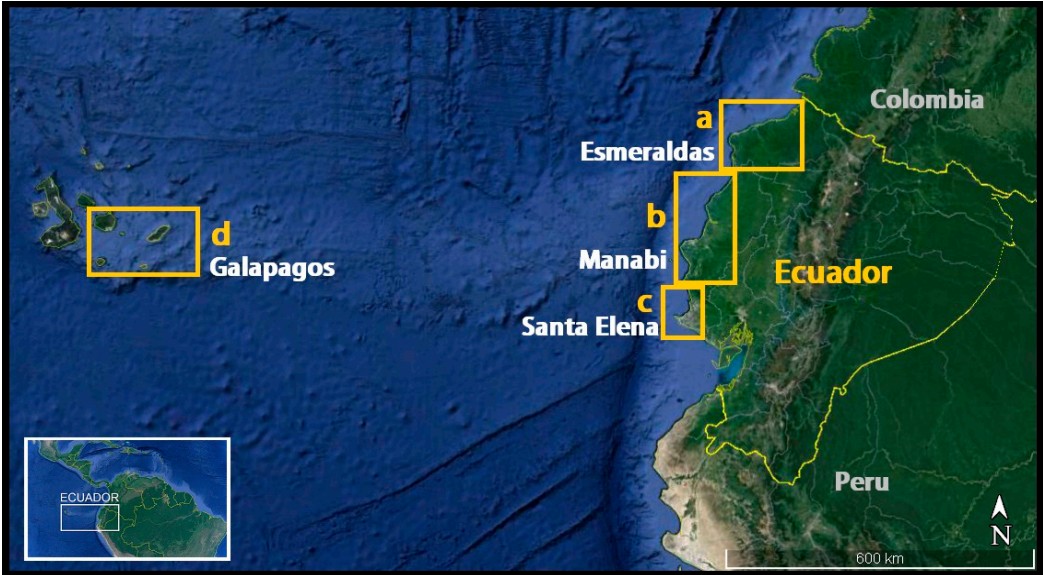

**Figure 1.** Location map showing the investigated sites in Ecuador.

The Galapagos province consists of 234 islands, islets and emerged rocks that give rise to an attractive and wide archipelago of great ecological relevance [23]. Sandy sectors, essentially constituted by pocket beaches, are composed by white, coral-rich sand with headlands and cliffs composed by black volcanic rocks, whose formation, emergence, and paleogeography are still highly uncertain [24]. The Galapagos Islands were declared World Heritage Natural Site by UNESCO in 1978 and, since that date, their reputation and eminence of being one of the last pristine natural paradises on Earth greatly grow up all around the world, especially because of their great biodiversity and, to a lesser extent, their geological heritage [25]. Galapagos presents one of the fastest growing economies in the world [23,26] and its urban areas are minimal and mainly present in four islands, i.e., Santa Cruz, San Cristóbal, Isabela and Floreana, which have a total population of 25,124 inhabitants.

From a physiographic point of view, the Ecuadorian coast is characterized by the presence of cliffs, beaches and estuaries. The most representative cliffs are found in the provinces of Esmeraldas between the beaches of Same and Escondida, and in the province of Manabí in the beaches of Punta de los Frailes, San Lorenzo and Los Frailes, the latter belongs to the Machalilla National Park [27,28].

The most important protected area in the Galapagos is the Galapagos National Park and, in the continental zone, there are the Machalilla National Park, the Galera San Francisco Marine Reserve, the Pacoche Coastal Marine Wildlife Reserve, the El Pelado Marine Reserve and the Puntilla de Santa Elena Coastal Marine Fauna Production Reserve.

**Table 1.** Location and main characteristics of investigated sites: Name, province, protection feature, beach typology, "D" scenic value and class.

| N° | Name | Province | Type | "D" | Class |
|---|---|---|---|---|---|
| 1 | Escondida *,(1) | Esmeraldas | Remote | 0.94 | I |
| 2 | Isla Portete | | Rural | 0.84 | II |
| 3 | Punta Galera *,(1) | | Rural | 0.57 | III |
| 4 | Africa | | Remote | 0.51 | III |
| 5 | Paufi | | Rural | 0.42 | III |
| 6 | Estero Platano *,(1) | | Village | 0.39 | IV |
| 7 | (Rocafuerte) | | Rural | 0.37 | IV |
| 8 | Same 2 | | Remota | 0.34 | IV |
| 9 | Las Palmas | | Urbana | 0.27 | IV |
| 10 | Sua | | Urbana | 0.27 | IV |
| 11 | San Francisco *,(1) | | Village | 0.27 | IV |
| 12 | Río Verde | | Village | 0.18 | IV |
| 13 | Same 1 | | Village | 0.09 | IV |
| 14 | Bocana del Lagarto | | Village | 0.03 | IV |
| 15 | Mompiche | | Village | 0.01 | IV |
| 16 | Las Peñas | | Village | −0.05 | V |
| 17 | Tonsupa | | Urbana | −0.20 | V |
| 18 | Atacames | | Urbana | −0.20 | V |
| 19 | Las Palmas | | Urbana | −0.27 | V |
| 20 | El Garrapatero *,(2) | Galapagos | Remote | 1.21 | I |
| 21 | Puerto Chino *,(2) | | Remote | 1.14 | I |
| 22 | Tortuga Bay *,(2) | | Remote | 1.13 | I |
| 23 | Mansa *,(2) | | Remote | 1.07 | I |
| 24 | Lobería *,(2) | | Remote | 0.99 | I |
| 25 | Punta Carola *,(2) | | Remote | 0.93 | I |
| 26 | Tijereta *,(2) | | Remote | 0.92 | I |
| 27 | Ratonera *,(2) | | Village | 0.87 | I |
| 28 | Estación *,(2) | | Village | 0.72 | II |
| 29 | Mann *,(2) | | Village | 0.52 | III |
| 30 | Los Alemanes *,(2) | | Village | 0.45 | III |
| 31 | Oro *,(2) | | Urban | −0.50 | V |
| 32 | Los Frailes *,(3) | Manabi | Remota | 1.17 | I |
| 33 | San José 2 *,(4) | | Remota | 1.00 | I |
| 34 | Cabuyal | | Remota | 0.94 | I |
| 35 | Punta Prieta | | Remota | 0.90 | I |
| 36 | Salango 2 *,(3) | | Remota | 0.88 | I |
| 37 | Tasaste | | Rural | 0.74 | II |
| 38 | Ayampe | | Village | 0.74 | II |
| 39 | San José *,(4) | | Rural | 0.71 | II |
| 40 | San Lorenzo *,(4) | | Village | 0.69 | II |
| 41 | La Tiñosa | | Rural | 0.65 | II |
| 42 | Don Juan | | Rural | 0.61 | III |
| 43 | Salango | | Village | 0.53 | III |
| 44 | San Clemente | | Rural | 0.50 | III |
| 45 | Sol | | Rural | 0.48 | III |
| 46 | Las Tunas | | Rural | 0.40 | III |
| 47 | Punta (del Fraile) | | Village | 0.24 | IV |
| 48 | Puerto Cayo | | Village | 0.23 | IV |
| 49 | Puerto Lopez | | Urbana | 0.22 | IV |
| 50 | Pedernales | | Urbana | 0.21 | IV |
| 51 | Canoa | | Village | 0.07 | IV |

**Table 1.** *Cont.*

| N° | Name | Province | Type | "D" | Class |
|---|---|---|---|---|---|
| **52** | San Mateo | | Village | 0.05 | IV |
| **53** | Crucita | | Urbana | 0.00 | IV |
| **54** | Santa Marianita | | Village | 0.00 | IV |
| **55** | Machalilla | Manabi | Village | −0.01 | V |
| **56** | San Vicente | | Urbana | −0.09 | V |
| **57** | Murciélago | | Urbana | −0.22 | V |
| **58** | Tarqui | | Urbana | −0.29 | V |
| **59** | Bahía de Caráquez | | Urbana | −0.57 | V |
| **60** | Rosada *,(5) | | Rural | 0.95 | I |
| **61** | Olon | | Village | 0.67 | II |
| **62** | Punta Carnero | | Urbana | 0.19 | IV |
| **63** | Puntilla de Santa Elena *,(6) | Santa Elena | Rural | 0.17 | IV |
| **64** | Ayangue *,(5) | | Village | 0.09 | IV |
| **65** | Montañita | | Urbana | −0.36 | V |
| **66** | Salinas Chipipe | | Urbana | −0.37 | V |
| **67** | Salinas San Lorenzo | | Urbana | −0.52 | V |

* Protected natural area, (1) Galera San Francisco Marine Reserve, (2) Galápagos National Park, (3) Machalilla National Park, (4) Pacoche Coastal Marine Wildlife Reserve, (5) Pelado Marine Reserve, (6) Puntilla de Santa Elena Coastal Marine Fauna Protection Reserve. Type beach: Urban = Beaches that have commerce and are freely accessible to the general public. Village = Beaches outside the urban environment, with a small population with organized community services on a small scale. Rural = Beach located outside the urban environment and difficult to access by public transport and generally without public service facilities. Remote = Beaches characterized by its difficult access; there is no public transport [18].

## 2.2. Methods

Several early studies were carried out on landscape assessments, e.g., [29–32], and they underlined the importance of limiting subjectivity, so that results "could be used in many planning and decision-making contexts". The above studies utilized different techniques, such as landscape assessment parameters, photographs, scenic uniqueness, best/worse scores from grid squares, public attitudes and perception, among others.

The methodology used in this paper was the result of an investigation financed by the British Council [33] subsequently published [1,16] and based on a checklist approach obtained by enquiring >1000 beach users chosen by random number tables in Malta, Turkey and the UK. Beach users were asked what was important for coastal scenic assessment, i.e., 'what are the essential parameters that make up a beautiful coastal scene' and, conversely, the 'coastal ugliness'. The results allowed the establishment of a checklist of 26 parameters (18 physical and 8 human, Table 2), which were evaluated from a low score (1), i.e., absence/poor quality, to a high score (5), i.e., excellent/outstanding (Tables 1 and 2). The 26 parameters were then assessed by a further group of beach users (>500 enquires carried out in the above-mentioned countries) to determine their relative importance, i.e., all parameters are NOT equal, some being more important than others. Further, to limit errors linked to subjective pronouncements and uncertainties inherited in assessment parameters, a Fuzzy Logic Assessment (FLA) [34] approach was used [16]. FLA represents a mathematical, analytical tool used when the complexity of the process in question is very high and there are no precise mathematical models to solve it, such as for highly non-linear processes. FLA has been used in many fields where subjectivity influences the achievement of accurate results, from financial systems to the remote sensing of cloud and ice cover. In the coastal scenic assessment, it was introduced to eliminate the possibility of the scenic value assessor (who ticks one box for each parameter) ticking the wrong attribute box due to uncertainty in the values shown [34], a jump of two attributes being extremely unlikely, e.g., checking an attribute 2 rather than 4 [16].

**Table 2.** Coastal scenic evaluation system. Physical and human parameters.

| No | | Parameters | Rating | | | | |
|---|---|---|---|---|---|---|---|
| | | | 1 | 2 | 3 | 4 | 5 |
| 1 | CLIFF | Height (H) | Absent | 5 m ≤ H < 30 m | 30 m ≤ H < 60 m | 60 m ≤ H < 90 m | H ≥ 90 m |
| 2 | | Slope | <45° | 45°–60° | 60°–75° | 75°–85° | circa vertical |
| 3 | | Features * | Absent | 1 | 2 | 3 | Many (>3) |
| 4 | | Type | Absent | Mud | Cobble/Boulder | Pebble/Gravel | Sand |
| 5 | BEACH FACE | Width (W) | Absent | W < 5 m or W >100 m | 5 m ≤ W < 25 m | 25 m ≤ W < 50 m | 50 m ≤ W ≤ 100 m |
| 6 | | Color | Absent | Dark | Dark tan | Light tan/bleached | White/gold |
| 7 | | Slope | Absent | <5° | 5°–10° | 10°–20° | 20°–45° |
| 8 | ROCKY SHORE | Extent | Absent | <5 m | 5–10 m | 10–20 m | >20 m |
| 9 | | Roughness | Absent | Distinctly jagged | Deeply pitted and/or irregular | Shallow pitted | Smooth |
| 10 | DUNES | | Absent | Remnants | Fore-dune | Secondary ridge | Several |
| 11 | VALLEY | | Absent | Dry valley | (<1 m) Stream | (1–4 m) Stream | River/limestone gorge |
| 12 | SKYLINE LANDFORM | | Not visible | Flat | Undulating | Highly undulating | Mountainous |
| 13 | TIDES | | Macro (>4 m) | - | Meso (2–4 m) | - | Micro (<2 m) |
| 14 | COASTAL LANDSCAPE FEATURES ** | | None | - | 2 | 3 | >3 |
| 15 | VISTAS | | Open on one side | Open on two sides | - | Open on three sides | Open on four sides |
| 16 | WATER COLOUR & CLARITY | | Muddy brown/grey | Milky blue/green/opaque | Green/grey/blue | Clear blue//dark blue | Very clear turquoise |
| 17 | NATURAL VEGETATION COVER | | Bare (<10% vegetation only) | Scrub/garigue (marron/gorse, bramble, etc.) | Wetlands/meadow | Coppices, marquis (±mature trees) | Varity of mature trees/mature natural cover |
| 18 | VEGETATION DEBRIS1 | | Continuous (>50 cm high) | Full strand line | Single accumulation | Few scattered items | None |
| 19 | NOISE DISTURBANCE | | Intolerable | Tolerable | - | Little | None |
| 20 | LITTER | | Continuous accumulations | Full strand line | Single accumulation | Few scattered items | Virtually absent |
| 21 | SEWAGE DISCHARGE EVIDENCE | | Sewage evidence | - | Same evidence (1–3 items) | - | No evidence of sewage |
| 22 | NON_BUILT ENVIRONMENT | | None | - | Hedgerow/terracing/monoculture | - | Field mixed cultivation ± trees/natural |
| 23 | BUILT ENVIRONMENT *** | | Heavy Industry | Heavy tourism and/or urban | Light tourism and/or urban and/or sensitive | Sensitive tourism and/or urban | Historic and/or none |
| 24 | ACCESS TYPE | | No buffer zone/heavy traffic | No buffer zone/light traffic | - | Parking lot visible from coastal area | Parking lot not visible from coastal area |
| 25 | SKYLINE | | Very unattractive | - | Sensitively designed high/low | Very sensitively designed | Natural/historic features |
| 26 | UTILITIES **** | | >3 | 3 | 2 | 1 | None |

The left side of the table has a vertical spanning label "Physical" for rows 1–18 and "Human" for rows 19–26.

* Cliff Special Features: indentation, banding, folding, screes, irregular profile; ** Coastal Landscape Features: Peninsulas, rock ridges, irregular headlands, arches, windows, caves, waterfalls, deltas, lagoons, islands, stacks, estuaries, reefs, fauna, embayment, tombola, etc.; *** Built Environment: Caravans will come under Tourism, Grading 2: Large intensive caravan site, Grading 3: Light, but still intensive caravan sites, Grading 4: Sensitively designed caravan sites.; **** Utilities: Power lines, pipelines, street lamps, groins, seawalls, revetments.

The employed algorithm involved both weighting and fuzzy logic values and included all of the above, enabling a Scenic Evaluation Value ("D") to be obtained indicating the 'beauty' of any particular site. "D" is calculated from membership degree versus attributes graph, and is the total area under the curve given from the following equation:

$$D = (-2a \cdot A_{1-2}) + (-1a \cdot A_{2-3}) + (1a \cdot A_{3-4}) + (2a \cdot A_{4-5}) \qquad (1)$$

Assessment matrices were calculated where $A_{1-2}$ is equal to total area under the curve between attributes 1 and 2. Similarly, areas under the curve may be calculated for $A_{1-2}$, $A_{2-3}$, $A_{3-4}$, $A_{4-5}$.

"D" classifies coastal scenery sites into five distinct classes, whose limits coincide with clearly identifiable cut-off points (Table 1), from Class I (D ≥ 0.85; extremely attractive natural sites), Class II (0.85 < D ≥ 0.65), Class III (0.65 < D ≥ 0.4), Class IV (0.4 < D ≥ 0), to Class V (D < 0; very unattractive, intensively developed urban sites, Table 1). Classes I and V occur within the top 85th percentile and lowest 15th, respectively [16]. The testing break points for Gaussian distributions (0.05 level) conformed with normality, indicating *study unbiasedness* [1,16], and this has been confirmed by assessments in many countries, e.g., UK, Turkey, Croatia, Bosnia, Malta, Portugal, Tunisia, Cyprus, Japan, China, Pakistan, eastern USA, several Pacific islands, New Zealand. Normality tests using chi-square and Kolmogorov–Smirnov tests have been performed at the 5% significance.

In past decade, >4000 scenic assessments have been carried out in Australia, Brazil, Colombia, China, Croatia, Cuba, Fiji, Japan, Morocco, New Zealand, Pakistan, Portugal, Spain, USA, etc., and these breakpoint values have been found to be constant [16,20,35–39]. For this paper, scenic evaluations were carried out in situ in February 2018, between 10h00 and 18h00., when good sunshine conditions are observed, and other information was also gathered, such as site location in natural areas, and tourist development typologies (Table 1) [18,40].

All sites underwent an evaluation matrix and the results were presented as histograms, weighted average of attributes and membership degree of attributes. The histograms provided visual summaries for all 26 parameters and were very useful for the immediate evaluation of high and low scoring attributes [13,16,35,41]. The weighted averages delineated the relative comparisons of physical and human parameters and the degree of membership versus the attribute curve, presenting a general scenic evaluation where the interpretation of the curve is based on the slope.

## 3. Results

In this study, scenic evaluation scores for 67 sites (55 in the mainland and 12 in the Galapagos Islands) were produced according to the described methodology. Assessment matrices for three investigated sites belonging to Class I, III and V were calculated and showed as histograms, weighted average and membership degree curves. They give a visual state of scores and trends of physical and anthropic parameters and make the interpretation of the results easier. Indeed, histograms enabled immediate visual assessment of the 26 attributes scores (Figure 2) while weighted averages enabled a visual comparison of physical and human parameters (Figure 3). Membership degree *vs.* attribute curve gave an overall scenic assessment reflected by its skew: A curve skewed to the right reflected high scenic qualities due to low scoring on attributes 1 and 2, and *vice versa* for a left-hand skewed curve (Figure 4).

For example, El Garrapatero (Santa Cruz island; "D": 1.21) and Los Frailes (Manabi; "D": 1.17), which are natural beaches located in protected areas, showed high scores in physical parameters, e.g., beach and water color, highly undulating landform and natural cover, a 50-m-high cliff at Los Frailes and outstanding biodiversity in Garrapatero, and very low impact related to human pressure (Figures 2–5). The sites of Mann (Galapagos; "D": 0.52) and Punta Galera (Esmeraldas; "D": 0.57) are respectively situated in rural and village areas, presenting high values of natural parameters (rocky shore, high vegetation cover with mature trees, clear water, etc.) and intermediate scores at human components due to the proximity of urban developments. Lastly, Atacames (Esmeraldas; "D":

−0.20) and Montañita (Santa Elena; "D": −0.36) are both urban beaches with low scores at natural parameters and were deeply affected by human activities related to intensively built environment, utilities, unattractive skyline and litter [16,20].

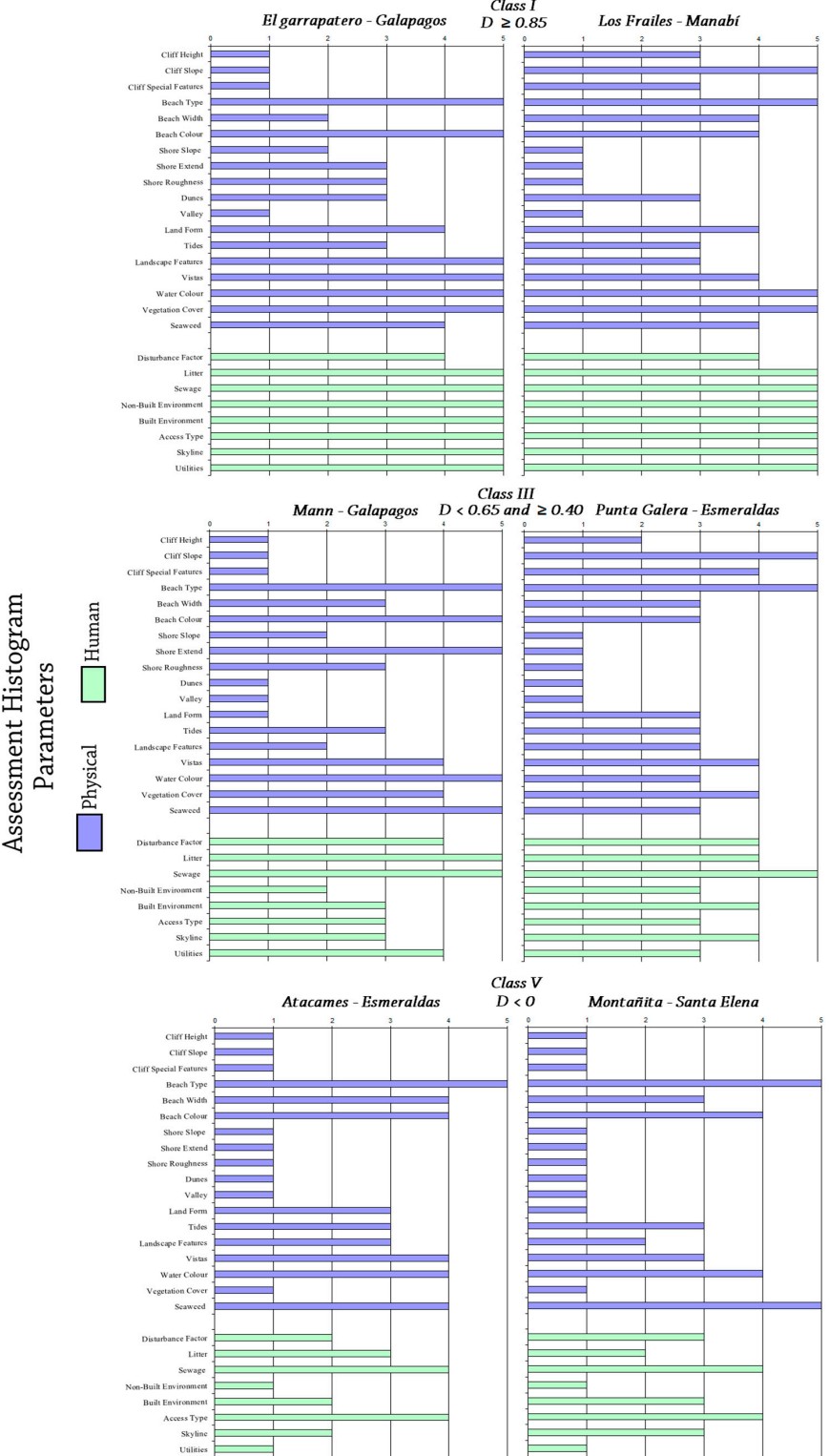

**Figure 2.** Scenic evaluation rating histograms for El Garrapatero and Los Frailes (Galapagos and Manabi provinces, Class I) Mann and Punta Galera (Galapagos and Esmeraldas, Class III) and Atacames and Montañita (Esmeraldas and Santa Elena, Class V).

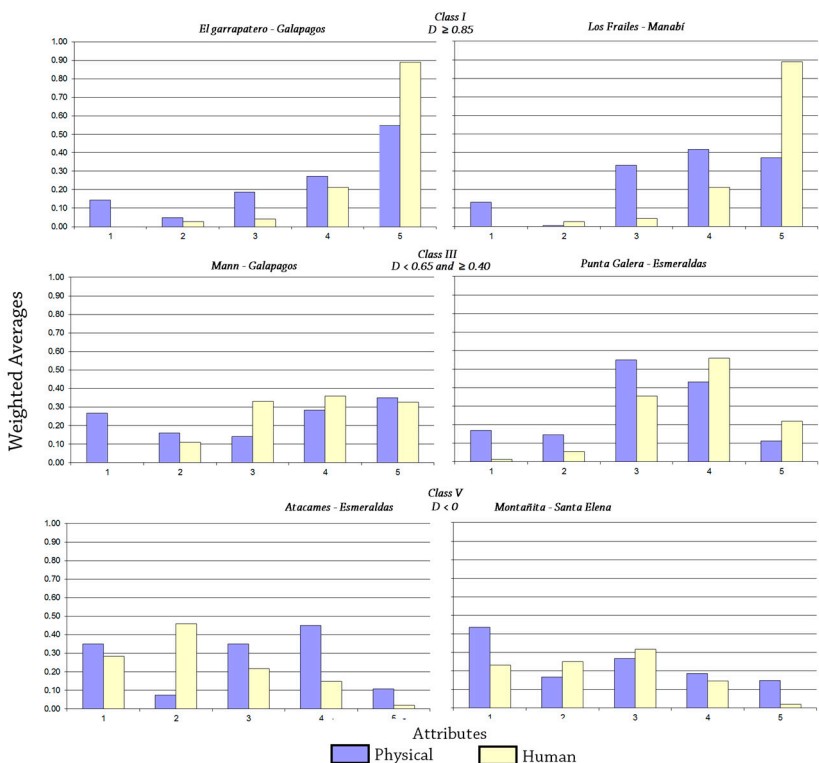

**Figure 3.** Weighted attributes for El Garrapatero and Los Frailes (Galapagos and Manabi provinces, Class I) Mann and Punta Galera (Galapagos and Esmeraldas provinces, Class III) and Atacames and Montañita (Esmeraldas and Santa Elena provinces, Class V).

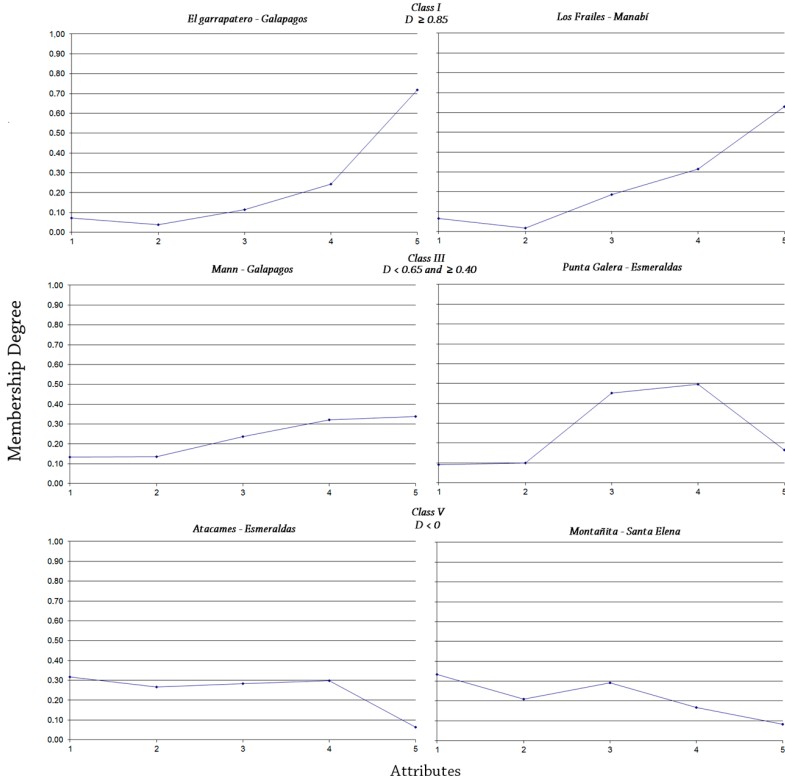

**Figure 4.** Membership degree *vs.* attribute curve for El Garrapatero and Los Frailes (Galapagos and Manabi provinces, Class I) Mann and Punta Galera (Galapagos and Esmeraldas provinces, Class III) and Atacames and Montañita (Esmeraldas and Santa Elena provinces, Class V).

The 67 investigated sites belonged to five classes, from Class I (top grade scenery) to Class V (very poor scenic value) and the analysis of the D value gave the following results (Figure 5).

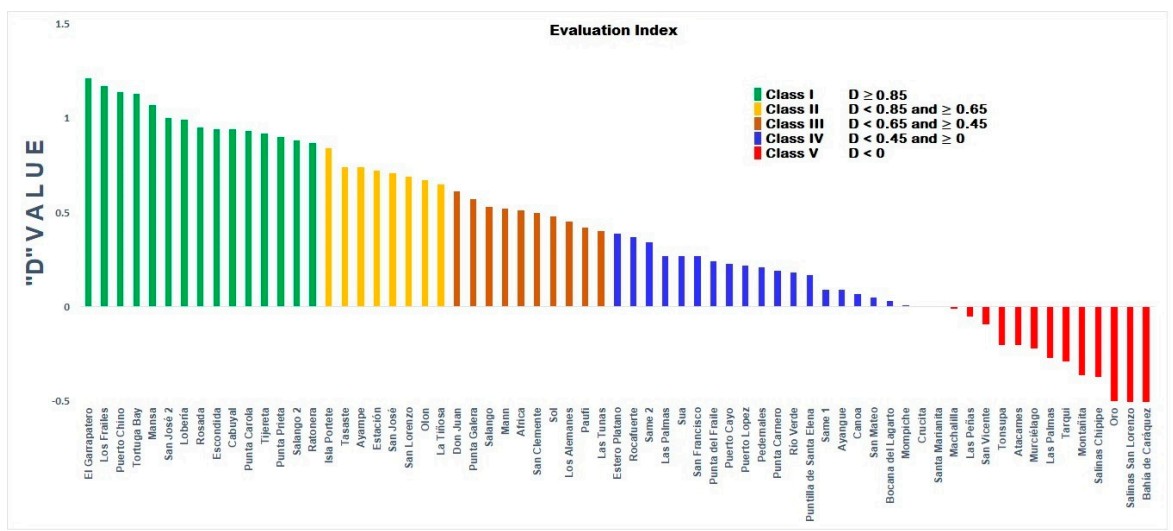

**Figure 5.** Evaluation index curve for the 67 sites investigated, 55 in the continental Ecuador and 12 in the Galapagos Islands.

### 3.1. Class I

These are sites with a value of "D" ≥ 0.85, very attractive beaches with very high landscape values (Figure 6). In total, 15 out of 67 beaches belonged to this class and 53% of them were located in the Galapagos Islands. About 90% of Class I beaches were remote, i.e., accessible by boat or by walking for 300 m or more on a country truck. Overall, 60% of Class I beaches were located in protected areas.

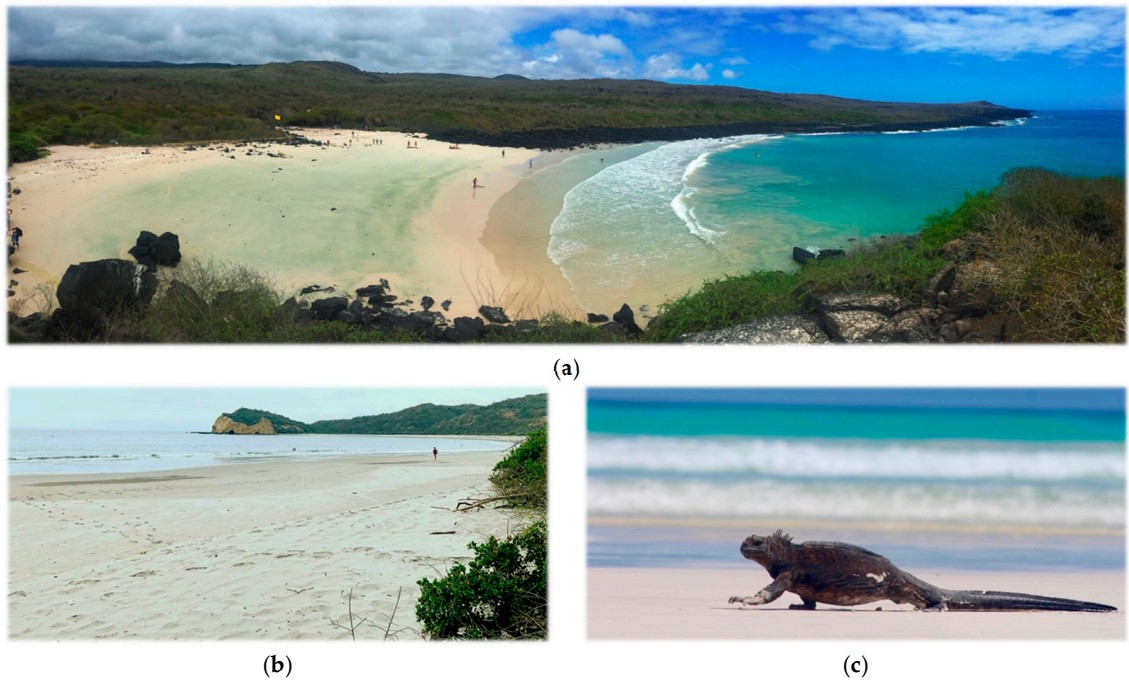

**Figure 6.** Class I, extremely attractive natural beaches with a very high landscape value: (**a**) Puerto Chino beach, San Cristobal island, Galapagos; (**b**) Los Frailes beach, Machalilla National Park; (**c**) Fauna on Garrapatero beach, Santa Cruz island, Galapagos. Fauna is a relevant variable in the assessment parameter, i.e., no. 14 "Coastal Landscape Features" (Table 2).

They may be adjacent to towns or rural areas, rarely are close to urban areas and they are not served by public transport. An example of this class is given by two beaches located in protected areas under the figure of National Park, i.e., "El Garrapatero" and "Los Frailes" (Figure 6). The scores on anthropogenic parameters were high and symmetrical, there was no evidence of beach litter, noise disturbance or beach facilities.

### 3.2. Class II

They are rural and village sites constituted by attractive beaches with high scenic values ($0.65 \leq$ "D" $< 0.85$); eight beaches belonged to this class, five of them were located in the province of Manabí and one in each one of the provinces of Esmeraldas, Santa Elena and Galapagos (Figure 7), within the Galapagos National Park.

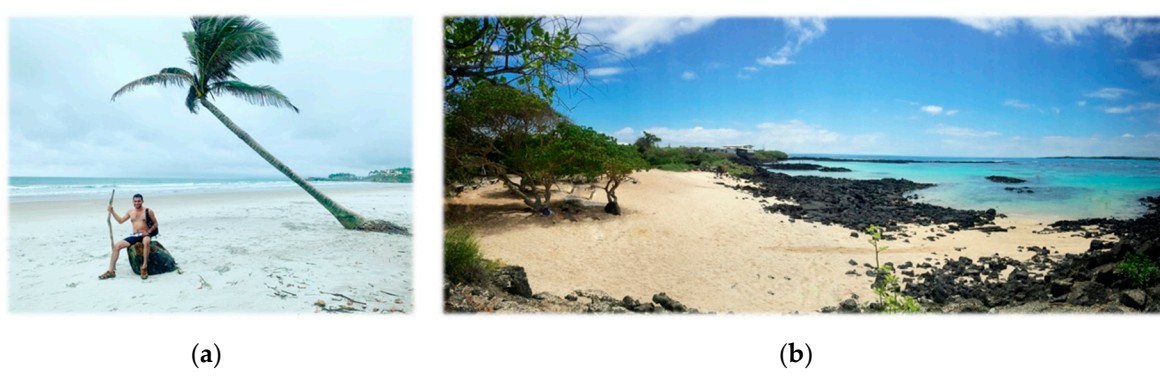

(**a**)　　　　　　　　　　　　　　　　　　　　　　　　　　(**b**)

**Figure 7.** Class II, attractive natural beaches with high landscape value: (**a**) Isla Portete beach, Esmeraldas province; (**b**) Estacion beach, Santa Cruz island, Galapagos.

### 3.3. Class III

Sites with "D" value $\geq 0.4$ and $< 0.65$; 10 sites were classified in this class, most of them were observed in Manabí (5), Esmeraldas (3) (e.g., Punta Galera beach, Esmeraldas province, Figure 8a) and Galapagos (2) (e.g., Mann beach, San Cristobal island, Galapagos, Figure 8b).

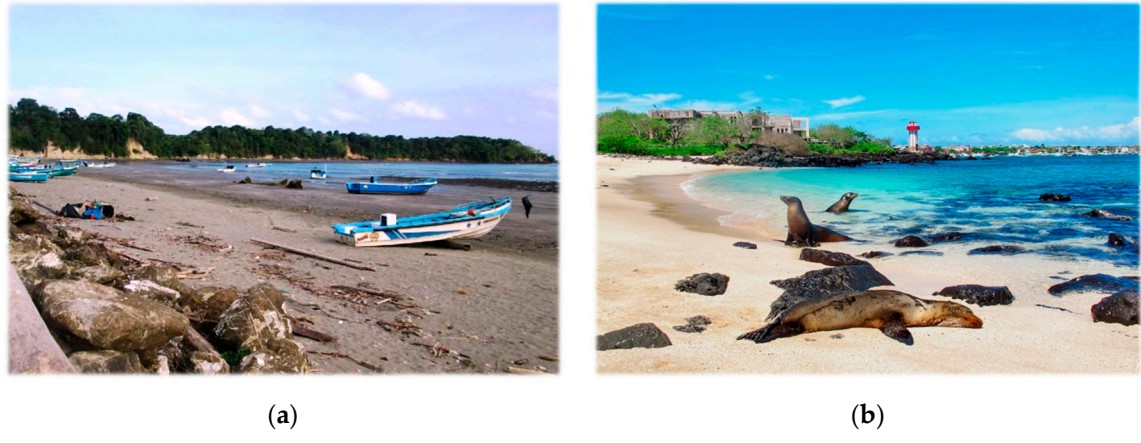

(**a**)　　　　　　　　　　　　　　　　　　　　　　　　　　(**b**)

**Figure 8.** Class III, essentially rural or village beaches with intermediate scores at natural and human parameters: (**a**) Punta Galera beach, Esmeraldas province; (**b**) Mann beach, San Cristobal island, Galapagos.

The anthropic influence on the beaches belonging to this class in the Galapagos National Park was high and natural aspects such as *Cliff*, *Dunes*, *Valley* and *Skyline* acquired low scores. Both sites in Galapagos Islands, Estacion and Mann, located, respectively, at the village of Puerto Ayora (Santa Cruz island) and Puerto Baquerizo (San Cristobal island), showed high natural scenic values as turquoise

water, mature vegetation cover and outstanding fauna (Figure 8) but were characterized by lower scores at human parameters as *Built environment*, *Access type* and *Skyline* (Table 2). Beaches of this class within the park were located in village areas and towns that reflected an organized service structure but at a small scale, such as schools, churches, shops and public or private transportation, aspects that are reflected in the parameters *Non-Built Environment* and *Built Environment* (Table 2).

## 3.4. Class IV

The largest number of sites evaluated was classified in this category, 21 out of 67, with a "D" value between 0 and 0.4. Beaches were essentially constituted by village (12), urban (6), rural (2) and remote (1) areas. Beaches with low natural parameters were observed in the continental zone in the province of Esmeraldas (10) (e.g., Estero Platano beach, Esmeraldas province, Figure 9a) and north of Manabí (8) (e.g., Pedernales beach, Manabí province, Figure 9b). These beaches were located near the mouths of rivers and mangroves' forests; as a result, *Water color & clarity* presented low scores.

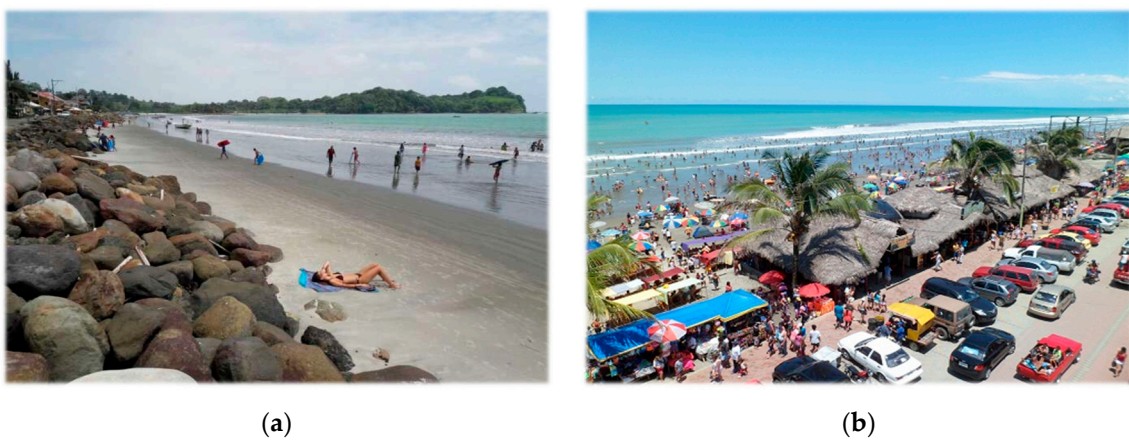

(**a**)          (**b**)

**Figure 9.** Class IV, mainly unattractive urban beaches with a low landscape: (**a**) Estero Platano beach, Esmeraldas province; (**b**) Pedernales beach, Manabí province.

The sites belonging to the province of Santa Elena presented low values at natural parameters such as *Rocky shore, Dunes, Valley* and *Vegetation cover* (Table 2). Low scores at anthropic parameters were essentially due to the characteristics of the built environment, access type, the presence of utilities (i.e., a rip-rap revetments and power lines) and skyline (i.e., the overall impact of human constructions).

## 3.5. Class V

Overall, 19% of the sites evaluated corresponded to this class (13 out of 67); they were essentially urban beaches (85%), i.e., areas freely accessible, with well-established public services, such as schools, banks and large commercial sites, which presented an intensive development and poor landscape values, i.e., "D" < 0.0. The provinces of Manabí and Santa Elena recorded 38% and Esmeraldas 21% of the sites evaluated in this category. Beaches in this class had poor values at natural parameters as *Cliff, Rock shore, Dunes, Valley, Landform, Landscape features* and *Vegetation cover* (Table 2). They were very crowded beaches all year round by a general public looking for fun (e.g., Atacames in Esmeraldas, Figure 10a; and Montañita in Santa Elena, Figure 10b). Anthropic parameters presented a high disturbance factor reflected by the presence of litter and the low scores recorded at *Built environment* and *Skyline* (Table 2).

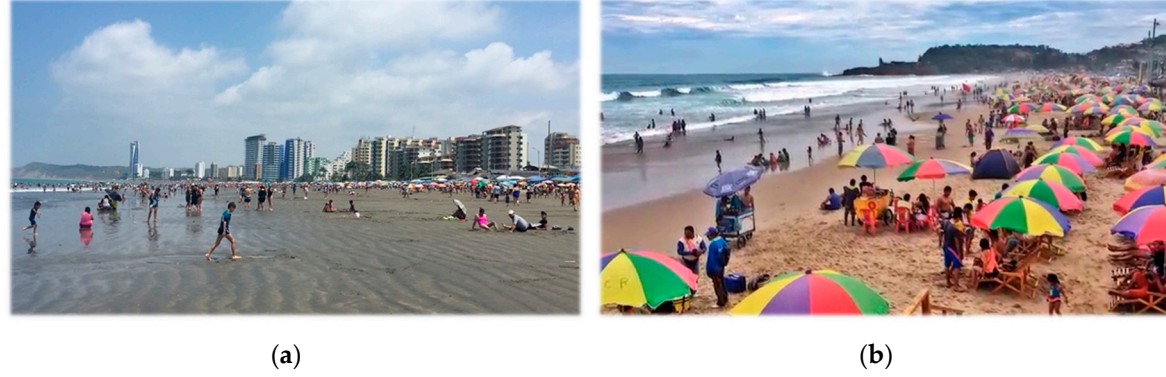

<div style="text-align:center">(<b>a</b>)        (<b>b</b>)</div>

**Figure 10.** Very unattractive urban beaches with intensive development, Class V: (**a**) Atacames beach, Esmeraldas province; (**b**) Montañita beach, Santa Elena province.

In Galapagos, a site was evaluated in this class, i.e., Playa Oro (Table 1), which is located in a protected area with the figure of National Park. Parameters that most determined this classification were human ones such as *Utilities*, *Skyline* and *Non-built environment* (Table 2). Natural parameters such as *Beach type*, *Beach color*, *Water color and clarity* and *Vegetation debris* had good scoring, but low scores were observed at *Dunes*, *Valley*, *Landform*, *Landscape features* and *Cliff* (Table 2).

## 4. Discussion

### 4.1. Human Impacts

The evaluation of the eight human parameters in the 67 study sites showed as beaches located within areas that have a protection figure presented lowest anthropic impacts reflected by good scores at all human parameters (Figure 3). When analyzing the presence of beach litter, sites with less presence of debris were those in protected areas especially in the Galapagos Islands National Park. This was due to the type of tourist who visits the beach (essentially international travelers) and their environmental consciousness [42], while the beaches with the lowest scores were the continental ones in the provinces of Esmeraldas and Santa Elena.

Beaches in urbanized areas, as Atacames and Montañita (Figure 10), were those that presented the worst evaluation in aspects such as *Built environment*, *Noise disturbance*, *Sewage discharge evidence*, *Access type* and *Utilities* (Table 2).

The assessment of 26 natural and human parameters carried out along the Ecuadorian coast allowed us to identify and characterize which variables could be managed in a better way to promote overall improvements of scenic value at many investigated sites. Regarding natural parameters, the formation of artificial dunes, beach nourishment, etc., are part of the few changes that can be carried out to upgrade their scenic quality. The main management actuations should be focused above all on anthropogenic aspects. At the Galapagos Islands, eight sites were ranked in Class I and this was partially due to the strict tourism regulations dictated by the National Park policy. On mainland, most of the investigated sites presented low scores at human parameters, mainly due to the absence of a buffer zone between the beach and the built environment and the lack of any kind of management. Indeed, the construction of human structures such as promenades, hotels, restaurants and other kinds of tourist developments based only on financial criteria (Figures 9b and 10), and the emplacement of groins, jetties and seawalls, considerably decreased physical parameters and associated landscape beauty, affecting at the same time coastal ecosystem services. Coastal erosion also had a negative effect on scenery since it reduced beach width (point 5, Table 2), such was the case of Mompiche (Esmeraldas; "D": 0.01), and at places induced the emplacement of different coastal protection structures (point 26, Table 2), e.g., Bocana del Lagarto (Esmeraldas, "D": 0.03), Estero Platano (Esmeraldas; "D": 0.39) (Figure 9a), San Francisco (Esmeraldas; "D": 0.27), San Clemente (Manabi; "D": 0.50), Montañita (Santa Elena; "D": −0.36). At the previously cited examples, the human impact on scenery was almost

irreversible. The absence of a buffer zone determined low scores at *Access type* and increased visual impact of the *Built* and *Non-built environment* (Table 2). Nevertheless, at many sites, low scores at human parameters were due to the presence of litter and sewage, and the presence of utilities such as beach kiosks and bars, restaurants, etc., directly placed on the back beach. If the presence of such structures would be regulated and reduced and cleaning operations implemented, such sites will clearly improve their scenic value possibly upgrading their class. For example, if the current litter score (2) at the Puntilla de Santa Elena (Santa Elena) will be improved to obtain a value of 4, the "D" value would increase from 0.17 (Class IV) to 0.46 (Class III). At Don Juan (Manabi), a rural beach with high natural scores, the establishment of periodic cleaning operations would upgrade the "D" value from 0.61 (Class III) to 0.73 (Class II). At the same place, several utilities were observed because management policies are very permissive and allow the presence of litter bins, beach bars, hammocks, etc.; however, if their number is reduced and their visual impact dissimulated, the site would change its score from 3 to 4 (point 26, Table 2) and, if litter presence is also reduced, the site would upgrade to Class I. Such is the case of Punta Galera (Esmeraldas), a sandy rural beach with high natural values, e.g., *Cliff*, *Shore platform*, *Vegetation cover*, etc. (Figure 2, Table 2); if the visual impact of utilities is reduced to a score of 4, the "D" value would upgrade from 0.57 (Class III) to 0.67 (Class II).

### 4.2. Management Issues

The Ministry of the Environment is acting with local governments in the coastal area and the Island region to join efforts to maintain a proper waste management plan. Programs for Solid Waste management have been implemented with the participation of the citizenry and educational institutions. These actions help to combat such aspects that represent a menace to biodiversity in coastal marine ecosystems, where hundreds of marine animals die annually due to the pollution linked to the presence of litter [43–45]. The awareness of tourists of not throw waste on the roads or beaches helps to avoid the deterioration of the natural environment.

The Governing Council of the Special Regime for Galapagos prohibited the trade, distribution, sale and delivery of disposable plastic bags. The island of Santa Cruz, the most populated of the Galapagos, has achieved the recovery of up to 45% of recyclable solid waste, the highest percentage in Ecuador, and this is reflected in the scenic quality of the beaches—88% of them have no litter [42].

At present, State tourism offices/departments and environment portfolios, through various projects with public and private organizations, seek to convert Ecuadorian beaches into quality tourism destinations by means of an integrated management with the active participation of all social actors, this way generating awareness and incrementing the respect for the environment [7,46,47]. Diagnostic actions are being carried out to improve the infrastructure and tourist services in coastal areas.

Tourism in Ecuador has been recognized as a national priority and considered a driving force within the fundamental axes of economic and social development. The Ministry of Tourism concentrates its management on five fundamental pillars that seek to position the country as an international tourism polo: security, quality, destination and products, connectivity and promotion. However, the recent budget reduction prevented reasonable results on these issues, as proven by the insecurity of tourists, the lack of maintenance of secondary roads, the decrease in tourism enterprises and the weak tourism promotion.

All the beaches studied in the Galapagos region were located in protected areas under the legal figures of *National Park* or *Marine Reserve*. Despite all the problems the Galapagos Islands have faced, especially linked to the presence of introduced species, they constitute one of the best conserved archipelagos around the world, and a world leader in the management of fragile ecosystems. The extraordinary flora and fauna, the geological characteristics, etc., have transformed this park into an important world center for scientific research and nature tourism [7,23].

At a continental level, countries such as Mexico and Colombia have based their development on mass 3S tourism, while Costa Rica, Cuba and Honduras are countries with sustainable and inclusive tourism. Colombia is focused more on adventure and quality tourism. Ecuador, on the

other hand, presents an almost unique potential in its tourism offer where the importance of nature and communities is enhanced. However, it still shows important limitations in its tourism policy such as the lack of institutions [7,48,49], quality of service, economic resources and joint actions with the private sector [50,51]. It is important to strengthen cultural and gastronomic aspects in order to potentiate tourism in the coastal profile, considering that the traditional aspects of the 3S (Sea, Sun and Sand) tourism in the continental Ecuador are not among the best in Latin America [52,53].

## 5. Conclusions

The coastal and insular area of Ecuador has innumerable sites of great tourist attraction that stand out for their varied culture and great biodiversity too, this representing for nearby urban and rural communities an opportunity for income generation and consequential economic development. Unfortunately, at the same time, limitations exist in the promotion of tourism and the potential of Public–Private Partnerships to strengthen the 3S tourism and compete with other countries in the region. To solve this issue, it is necessary to consolidate and promote the 3S tourism as a state policy; this will intensify public and private investment for the development of tourism and the construction of a favorable environment for local communities.

This study provided information on the scenic characteristics of the Ecuadorian coastal profile and the impact of tourism on the 67 beaches analyzed. The best beaches in Ecuador are located in the Galapagos Islands and, in the continental area, in Manabí and Santa Elena provinces. The Galapagos Islands, with their natural white sand beaches, surrounded by endemic flora and fauna and their black lava formations, have the largest number of Class I beaches. The province of Manabí, which shows high cliffs along almost all its coastal profile, has beaches with the best sand and water characteristics of the continental area. The province of Santa Elena is characterized by urban beaches of low scenic value, but natural beaches acquire medium scores. The beaches of Esmeraldas province have the lowest scenic quality due to the presence of dark sediments, murky waters and a flat landscape. They are generally visited by national tourists from the Andean and Amazon regions, while the tourists who visit the beaches of Manabí and Santa Elena come mostly from the large metropolises in the center–south of the country and have a low percentage of foreign tourists, although their number is higher than that of the beaches of Esmeraldas. Beaches located within the areas that have a protection figure show less anthropic impact. The tourists' provenience area has a great influence on the qualification of human parameters, too; it is evident that international guests come mostly to see natural values, and therefore are more careful of their conservation.

The results of this study constitute valuable information for future tourism management plans that can contribute to sound Integrated Coastal Zone Management (ICZM) actions. Coastal zone managers should focus their efforts on improving all the anthropogenic parameters investigated in this study. Correct management and action measures will allow to revert the negative human impacts on beaches' scenic value. A large percentage of the investigated beaches would upgrade their classification of 1 or even 2 classes if simple actions such as cleaning campaigns, maintenance and relocation of facilities will be implemented. Finally, it is important to apply the various coastal management policies that the country has planned in order to prevent the settlement of new industrial infrastructures that heavily affect the delicate balance of maritime–coastal ecosystems and coastal scenic beauty.

**Author Contributions:** Conceptualization, C.M.-R., G.A. and J.A.C.-R.; Formal analysis, C.M.-R., G.A., J.A.C.-R., A.M., C.M.B., and E.P.; Funding acquisition, C.M.-R. and G.A.; Investigation, C.M.-R., G.A., J.A.C.-R., A.M., C.M.B., and E.P.; Methodology, G.A. and A.M.; Project administration, C.M.-R., G.A. and J.A.C.-R.; Software, E.P.; Validation, C.M.B. and J.A.C.-R.; Writing—original draft, C.M.-R. and G.A.; Writing—review & editing, C.M.-R., G.A., J.A.C.-R., A.M., C.M.B., and E.P. All authors have read and agreed to the published version of the manuscript.

**Funding:** This research was funded by the Universidad de Cádiz (Spain), Escuela Superior Politécnica de Chimborazo (Ecuador) and GREEN AMAZON ECUADOR (Grant Number. 34323674) and it is a contribution to the PROPLAYAS network and the PAI RNM-328 Research Group of Junta de Andalucía (Spain).

**Acknowledgments:** The authors are grateful for the financial support of GREEN AMAZON ECUADOR and for the support of researchers from the Universidad de Cádiz (UCA), the Escuela Superior Politécnica de Chimborazo (ESPOCH), the University of Florence (Italy) and the Instituto Superior Tecnológico Universitario Oriente (ITSO). This paper is a contribution to the Andalusia PAI Research Group RNM-328 and the PROPLAYAS Network.

**Conflicts of Interest:** The authors declare no conflict of interest.

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
