# Peer review of "Coastal Scenic Evaluation of Continental Ecuador and Galapagos Islands: Human Impacts and Management Issues"

_jmse, doi:10.3390/jmse8060468_

Round 1

Reviewer 1 Report

Title: Coastal scenic evaluation of Continental Ecuador and 2 Galapagos Islands: human impacts and management 3 issues

Authors: Mestanza-Ramon et al.  

Manuscript ID: jmse-820171

General Comments:

Coastal area monitoring is very difficult to study accurately. So, the merging the data from select coastal destination (beautiful coastal scene and coastal ugliness) with 26 evaluation parameters may be a very important step in the Ecuador coastal scenic evaluation (i.e., water quality, landscape, security, etc). This paper addresses the challenging topic of scenic characteristics of the coastal profile and the impact of tourism on the beaches. Respect to the obtained results: The best beaches in Ecuador are in the Galapagos Archipelago and, in the other side, the beaches of Esmeraldas present lower scenic quality due to their proximity to the capital and are much visited. Overall, 22% (or 15) of the beaches investigated belonged to Class I, 12% (8) to Class II, 15% (10) to Class III, 31% (21) are in Class IV and 20% (13) in Class V.

This is an interesting subject but very theoretical and not an usual scientific work. The paper contains some interesting ideas but I find the structure of the paper rather confusing with too many general statements and too much repetition. The paper needs to be much more specific.

The main comments and considerations are:

In my opinion, the main shortcoming for this paper is lacking in novelty and the lack of comparison results and discussions. I suggest the authors carefully revise the manuscript and add some new experimental design (similar to figure 5, which is fine and clear.)

1.- Fundamental questions:

(i) I would like to know why this work is useful and how we gain new information from it and,

(ii) Respect to the proposed D value: Does it need to be evaluated under different criteria? How do you define the class limit: for example Class I D > 0.85 (i.e. see Figures 2 and 5)?

2.- The abstract must be revised since it is not especially informative.

3.- The introduction is very general about tourism statistical data (global and local). I think it must be rewritten.

4.- The results section should include other methods conducted in previous studies to demonstrate the utilities of the proposed method. Besides, the quantitative and qualitative comparison should be covered as well: This study provided limited information on the scenic characteristics of the Ecuadorian coastal profile and the impact of tourism on the 67 beaches analyzed.

5.- Conclusions: The conclusions are more like a summary of the results. This sections needs to be reconsidered. Maybe a little bit of more detail could be included.

In conclusion, I personally think that a lot of work was done but the current version of the manuscript need revisions.

Additionally, some figures and tables are not of good quality. Many errors in the format, text, references, etc. Extensive editing of English language and style is required

Author Response

Reviewer 1

Dear Reviewer

we carried out almost all changes you proposed and we attached a file with example of papers, books etc that used the method used in this paper.

Line

Comment

Action - Response

Introduction

--

I suggest the authors carefully revise the manuscript and add some new experimental design (similar to figure 5, which is fine and clear.)

We added new information.

---

--

I would like to know why this work is useful and how we gain new information from it and,

The paper is important because it provides relevant information to coastal managers and administrators in order to strengthen the management of the 3S tourism sector.

Further, it encourages future research on the presented topics.

---

20 – 33

The abstract must be revised since it is not especially informative.

The abstract has been extensively rewritten and the reviewer's comments were considered to make it more informative.

160 – 277

Respect to the proposed D value: Does it need to be evaluated under different criteria? How do you define the class limit: for example Class I D > 0.85 (i.e. see Figures 2 and 5)?

The fuzzy logic and the determination of "D" limits were explained.

37 – 81

The introduction is very general about tourism statistical data (global and local). I think it must be rewritten.

The introduction has been partially rewritten based on the reviewer’s recommendations trying to describe landscape and tourism in Ecuador.

160 – 277

The results section should include other methods conducted in previous studies to demonstrate the utilities of the proposed method. Besides, the quantitative and qualitative comparison should be covered as well: This study provided limited information on the scenic characteristics of the Ecuadorian coastal profile and the impact of tourism on the 67 beaches analyzed.

The classification used is the most common one in the determination of coastal landscape value and has been widely used in several countries. We upload with our revision a file with a list (not exhaustive) of papers, books, PhD thesis etc. existing on the topic and that used the Coastal Scenic Evaluation System.

Methods used in other studies were presented in the “Methods” Section:

“….such as landscape assessment parameters, photographs, scenic uniqueness, best/worse scores from grid squares, public attitudes and perception….”

369-405

Conclusions: The conclusions are more like a summary of the results. This sections needs to be reconsidered. Maybe a little bit of more detail could be included.

Conclusions were partially rewritten as requested.

170,174, 179, 197

Additionally, some figures and tables are not of good quality. Many errors in the format, text, references, etc.

The figures were edited and the quality was improved.

Reviewer 2 Report

Dear Authors,

your manuscript reports an extensive study on 67 beaches in Ecuador (both in mainland and Galapagos Islands), for which the coastal scenic value was defined considering 26 parameters of different nature, according to a procedure adopted for other sites and described in literature. A score was assigned to each parameter and results were analyzed using a fuzzy logic technique, in order to reduce the subjectivity of the analysis. A synthetic index (Scenic Evaluation Value) comprehensive of the scores of the different parameters was derived for each of the investigated sites, that were subdivided into different coastal scenery classes. The impacts of tourism and anthropogenic pressure were discussed, as well as the possible measures to improve the conditions of some sites and promote a sustainable tourism.

The work is interesting, especially considering the high number and the particular relevance of many of the the investigated sites. The paper is well organized and written. Overall, my general impression is that the paper is good, and I recommend it to the Editors for publication.

I add some suggestions that I hope would be helpful in order to further improve the paper:

  • A weighting factor was assigned to each of the 26 parameters. I would suggest spending some words to describe the weighting procedure and to indicate the parameters having the highest and lowest weights, indicative of their relative importance.
  • I would suggest adding some words to describe the Fuzzy Logic Assessment adopted to analyze the results.

Finally, I add some minor comments and observations:

  • The resolution of Figures 2, 3, 4 and 5 seems low and the text is not clearly readable. Probably it could depend on the conversion to pdf format. However, I would suggest a check.
  • Line 39: please, check the word “rich”.
  • Line 308: please, check the word “sores”.
  • Lines 350-357: the sense is clear, but the sentence seems to contain some errors. Please, check.

Kind regards.

Author Response

Reviewer 2

Line

Comment

Action

New Line

Introduction

39

please, check the word “rich”.

Done.

117 - 159

I would suggest spending some words to describe the weighting procedure and to indicate the parameters having the highest and lowest weights, indicative of their relative importance.

Done.

117 - 159

I would suggest adding some words to describe the Fuzzy Logic Assessment adopted to analyze the results.

The fuzzy logic .and the determination of "D" values were explained.

170,174, 179, 197

The resolution of Figures 2, 3, 4 and 5 seems low and the text is not clearly readable.

The figures were edited and the quality was improved.

308

please, check the word “sores”.

The word was corrected.

350-357

Lines 350-357: the sense is clear, but the sentence seems to contain some errors.

The paragraph was restructured.

Reviewer 3 Report

The manuscript provides an interesting method for touristic site ranking in order to improve the overall environmental quality. The method is simple and handy for management purpose. 

The overall quality of the text is quite good. It needs a moderate editing of the English language. Moreover the methodology description needs to be rewritten since Table 2 is discussed before Table 1. I've included some comments on the attached revised file of the manuscript.  

I would suggest moderate revision.

Kind regards

Author Response

Reviewer 3

Line

Comment

Action

Introduction

39

Modify writing “rich 1.5 billion that corresponded with”

The sentence was changed.

43

Modify writing “foreigners”

The word was modified.

44

Modify writing “Marked impostata da”

We are sorry we did not understand the question.

44

Modify writing “GDP” The first time you use an acronym, you should also write the full meaning

Done.

115

This Table was never quoted in the text before. Please, provide a full description of it and the meaning and source of the "D" and "Class" attributes

Table 1, was cited in the text.

129

This should be Table 1

The order of the tables was changed.

149

This should be table 2

The order of the tables was changed.

Round 2

Reviewer 1 Report

To Authors & Editors,

After reviewing in detail the answers and the amendments carried out in the manuscript, I consider that authors have successfully cover all my comments and suggestions. So, the revised version of the paper can be now accepted for publication with minor revisions (english style, tables format, figure resolutions, etc...).